# Protocol for developing a Consolidated Checklist for Reporting Mixed Methods Research (CORMIX) using modified Delphi

Myriam Jaam[1‡], Ahmed Awaisu[1☾], Derek Stewart[1☾], Banan Mukhalalati[1☾], Ahsan Sethi[2☾], Marwa Elshazly[1☾], Abrar Abdelrahman[1☾], Muhammad Abdul Hadi[1‡*]

**1** Department of Clinical Pharmacy and Practice, College of Pharmacy, QU Health Sector, Qatar University, Doha, Qatar, **2** QU Health Sector, Qatar University, Doha, Qatar

☾ These authors contributed equally to this work.
‡ MJ and MAH are Joint Senior Authors.
* mabdulhadi@qu.edu.qa

## Abstract

### Background

Mixed methods research has gained popularity across healthcare and social sciences. However, inconsistent, and inadequate reporting remains a concern, hindering the assessment and utilization of mixed methods studies. Comprehensive easy-to-use mixed methods reporting standards are lacking. This protocol outlines the development of the COnsolidated Checklist for Reporting Mixed Methods Research (CORMIX) to address this gap.

### Methods

Following the EQUATOR Network guidelines, the protocol involves conducting a scoping review to identify potential checklist items from current reporting guidelines and existing literature. This will be followed by executing a modified Delphi with a panel of experts in mixed methods research for further refinement through consensus. The modified Delphi will include ≥40 purposively selected experts who have extensively utilized and published mixed methods research. A questionnaire compiling literature-derived reporting items will be validated and distributed in successive rounds until consensus is reached on all items. Consensus is defined as ≥ 85% of experts agreeing on the inclusion of an item. Following the finalization of the checklist, a user guide and justification document will be completed.

### Discussion

This rigorous development process conforms with EQUATOR Network standards for health reporting tools. CORMIX has the potential to enhance the quality, consistency, and transparency of mixed methods reporting. Clear reporting is essential

**Data availability statement:** Currently no data is available as this is a protocol. However, upon completion of the study the data will be made available.

**Funding:** The author(s) received no specific funding for this work.

**Competing interests:** The authors have declared that no competing interests exist.

for assessing validity, replicating studies, and enabling synthesis and application of mixed methods evidence. The protocol is currently registered with the EQUATOR network: CORMIX – COnsolidated Checklist for Reporting Mixed-Methods Research.

## Introduction

Mixed methods research has gained huge popularity within the fields of healthcare, education, and social sciences in the past few decades, as it provides a deeper understanding of complex phenomena by combining the statistical robustness of quantitative data with the rigour and richness of qualitative data, thus enhancing research findings [1–4]. Combining qualitative and quantitative data, in mixed methods research, capitalizes on the strengths of each - qualitative data offers in-depth insight to complement the generalizability of quantitative findings [1,2]. The depth provided by qualitative data can help explain relationships revealed in quantitative data. Likewise, quantitative data provides generalization to complement the contextual insights from qualitative inquiry [1,2]. Using mixed methods enables researchers to develop a more well-rounded understanding of a subject. In fact, the number of papers related to mixed methods research has tripled between 2010 and 2020 [5]. By integrating qualitative and quantitative methods within a single study, mixed methods research allows researchers to answer complex research questions, and provide a complete picture of the research topic [1,6].

However, concerns about transparency and suboptimal reporting of mixed methods research have been documented in the literature [7–13]. Some researchers fail to effectively report the integration of qualitative and quantitative data or provide a limited integration at the analysis and interpretation stage [7,10,11,13–16]. Other studies have noted deficiencies in how key methodological components of qualitative and quantitative research are reported in mixed methods studies. Important details regarding sample selection, sampling procedures, study context, and data-gathering procedures were frequently missing [8,12,16]. Furthermore, studies have indicated that the use of mixed methods research has not been well justified [1,14]. These reported limitations highlight that researchers merely use mixed methods as a mean to collect large volumes of qualitative and quantitative data with limited integration. Hence, the full potential of mixed methods is not realized; the qualitative and quantitative components end up seeming disconnected instead of offering complimentary insights, resulting in a less comprehensive understanding of the research topic [9,10,16].

As highlighted by the EQUATOR Network (Enhancing the QUAlity and Transparency Of health Research), clear and complete reporting of research is critical for several reasons. It allows editors, reviewers, and readers to fully understand what was done and enables others to replicate the study. Studies have shown that when journals endorse reporting guidelines, there is an improvement in adherence, transparency, and the overall impact of research findings [17]. Incomplete reporting can distort the scientific literature and hamper the judicious utilization of the research

[18]. Inadequate reporting impacts readers' understanding of the purpose and benefits of using mixed methods research, which hinders future researchers from designing their own mixed methods research studies, and limits policymakers' ability to make informed decisions based on comprehensive evidence [4,7,19].

Unlike guidelines for purely quantitative (e.g., CONSORT for randomized trials) [20]or qualitative studies (e.g., COREQ) [21], mixed methods reporting guidelines need to address the complexities of combining these different approaches. By establishing clear standards for reporting, these guidelines can enhance the evaluation, synthesis, and practical application of mixed methods research. This is particularly relevant for informing policy decisions and practice [2].

While robust guidelines exist for mixed methods research reporting, there remains a need for a unified, standalone checklist that streamlines the reporting process and provides detailed operational guidance across disciplines. [10,22,23]. For example, the Good Reporting of A Mixed Methods Study (GRAMMS) tool outlines core reporting items for mixed methods research [22]. It provides a general reporting criteria but lacks detailed guidance on reporting integration of qualitative and quantitative components [24]. More recently, Levitt et al. (2018) introduced the Mixed Methods Article Reporting Standards (MMARS) as part of the broader Journal Article Reporting Standards for Qualitative Research (JARS-Qual) [25]. The MMARS provides guidelines for reporting mixed methods research in psychology and related fields. While more comprehensive than GRAMMS, it was primarily developed for psychology and related fields, while mixed methods research spans multiple disciplines including healthcare, education, and social sciences. Additionally, users must cross-reference between multiple documents (JARS-Qual [25] and JARS-Quant [26]), which can complicate the reporting process and compromise adhering to the tool. A standalone, unified checklist could enhance usability and hence, improve reporting quality.. This is not a critique of MMARS's comprehensiveness, but rather acknowledges the practical challenges researchers face in implementing existing guidelines and the need for tools that can enhance accessibility and usability across fields. Standards like SRQR [27], COREQ [21], JARS-Quant[26], and JARS-Qual [25] provide guidelines for reporting qualitative, and quantitative research independently. There is currently no comprehensive, easy-to-use checklist specifically tailored to the unique complexities of mixed methods research studies that can be used across different fields.A recent editorial published in the Journal of Mixed Methods Research highlighted the urgent need to develop reporting guidelines for mixed methods research [28].

It is important to distinguish between quality assessment tools, which evaluate a study's methodological strength, and reporting guidelines, which provide a framework for authors to clearly communicate their research process and findings [29,30]. While methodological quality is crucial, our current work specifically addresses the standards and practices for comprehensive reporting of mixed methods research. To address this need, this is a protocol to develop a comprehensive widely applicable mixed methods reporting tool that is adaptable across various disciplines, called Consolidated Checklist for Reporting Mixed Methods Research (CORMIX), using the modified Delphi method. The modified Delphi technique is a structured, iterative communication method that involves multiple rounds of data collection and feedback from a panel of experts. It can be used to achieve consensus on a given topic or develop content like a reporting checklist. In this case, we will conduct an initial literature search to identify potential checklist items, and then use the modified Delphi process to refine the checklist through expert feedback and consensus. [31].

## Methods

The reporting checklist will be developed in accordance with the guidelines developed by Moher et al. [29] and endorsed by EQUATOR [18]. These guidelines have been widely adopted in health research, forming the basis for reporting checklists like CONSORT [20] and PRISMA [32] which are considered gold standards. The steps outlined, such as conducting systematic reviews to identify current reporting gaps, obtaining feedback from relevant stakeholders, and iteratively refining the checklist, help ensure a thorough development process. The steps recommended by Moher et al are presented in the context of this project in Fig 1. Step 3: Obtaining funding for the guideline initiative was eliminated from Fig 1 as no funding is associated with this project.

## Registration and ethical approval

The reporting guideline under development is now registered on the website of EQUATOR Network: CORMIX – COnsolidated Checklist for Reporting Mixed Methods Research [33].

Informed written consent will be obtained from experts participating in the Delphi phase through email before completing the questionnaire.

This project received ethical approval from Qatar University Institutional Review Board (QU-IRB). Reference number: QU-IRB 1916-E/23.

## The research team

The core research team consists of eight authors who have a diverse background in conducting mixed methods research and are interested in evidence-based medicine and quality improvement. A short biography of each of the core research team is attached as a supplementary file. The research team will work collaboratively in identifying relevant CORMIX items through a literature search and arranging and rephrasing the items into a suitable checklist. Then, they will execute a modified Delphi with a panel of experts, conduct data analysis, and finalise the checklist. They will also work through drafting the Explanation and Elaboration E&E document and through publication,

## Initial phase

**Identifying the need for a guideline.** An initial search was conducted in different databases and the EQUATOR network website to identify if there was an existing checklist related to mixed methods research. Identified tools were the MMAT (Mixed Method Appraisal Tool), MMARS, GRAMMS, QATSDD (Quality Assessment Tool of Studies with Diverse Design) and its update QuADS (Quality Assessment with Diverse Studies), MMR-RHS (Standards for Mixed Methods Reporting in Rehabilitation Health Sciences Research), ASSESS tool (a comprehenSive tool to Support rEporting and critical appraiSal of qualitative, quantitative, and mixed methods implementation reSearch outcomes) [22,25,34–38].

While these tools have made significant contributions to the field, they each have different focuses and limitations with regard to reporting. For instance, the MMAT, QATSDD, and QuADS primarily emphasize quality assessment rather than comprehensive reporting guidance. The MMARS, provides guidelines for psychology and related fields, but requires cross-referencing with other reporting tools, which can complicate its application. GRAMMS offers important initial criteria but may lack the detail needed for comprehensive reporting across all aspects of mixed methods studies. The ASSESS tool provides a comprehensive framework for reporting and critically appraising implementation research across quantitative, qualitative, and mixed methods studies. The tool's focus on implementation outcomes may not align well with mixed methods studies that have a broader focus or different primary objectives. Additionally, the sections pertaining to mixed methods are originally sourced from the MMAT which is a methodological quality assessment tool rather than a reporting tool. The MMR-RHS checklist, while well designed for rehabilitation and health sciences, may not fully capture the nuances of mixed methods research across different disciplines. For instance, in social sciences, philosophical stances (e.g., pragmatism, constructivism, transformative paradigms) often fundamentally guide research design, yet the MMR-RHS doesn't explicitly prompt reporting of these considerations. This limitation is also seen in other tools like ASSESS, despite philosophical and paradigmatic considerations being widely discussed in mixed methods literature. Given these gaps in existing reporting frameworks, our study will use expert consensus to evaluate the role and importance of reporting such considerations in mixed methods research.

While these tools have been developed with input from experts and authors in the field, there remains a need for a more comprehensive, standalone reporting guideline that addresses the full spectrum of mixed methods research complexities and is based on a robust, systematic methodology. Multiple reviews highlighted the inconsistencies in reporting mixed methods research, emphasising the need for mixed methods research reporting guidelines [10,22,23,39–41].

 

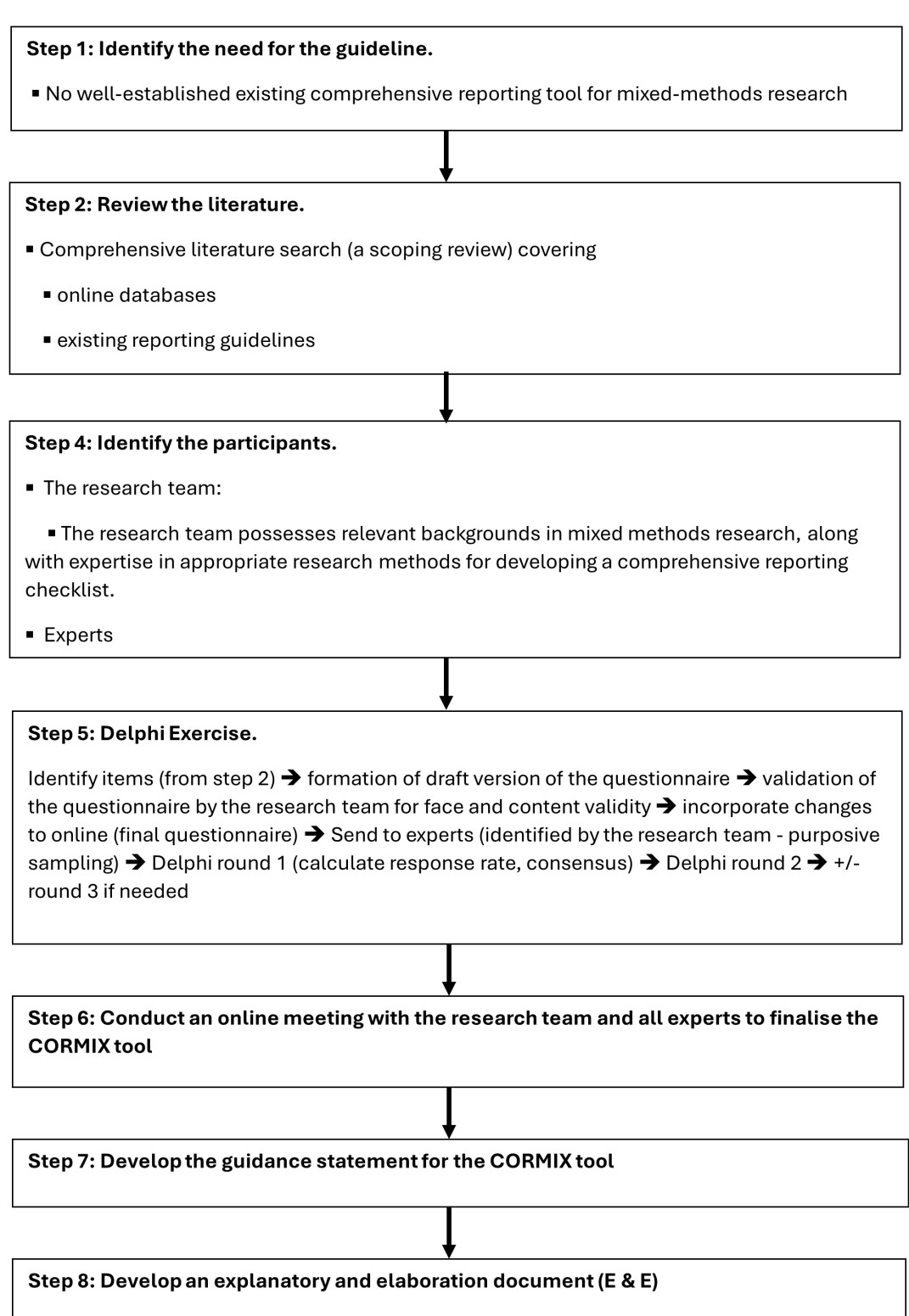

**Step 1: Identify the need for the guideline.**

■ No well-established existing comprehensive reporting tool for mixed-methods research

**Step 2: Review the literature.**

■ Comprehensive literature search (a scoping review) covering

   ■ online databases

   ■ existing reporting guidelines

**Step 4: Identify the participants.**

■ The research team:

   ■ The research team possesses relevant backgrounds in mixed methods research, along with expertise in appropriate research methods for developing a comprehensive reporting checklist.

■ Experts

**Step 5: Delphi Exercise.**

Identify items (from step 2) ➜ formation of draft version of the questionnaire ➜ validation of the questionnaire by the research team for face and content validity ➜ incorporate changes to online (final questionnaire) ➜ Send to experts (identified by the research team - purposive sampling) ➜ Delphi round 1 (calculate response rate, consensus) ➜ Delphi round 2 ➜ +/- round 3 if needed

**Step 6: Conduct an online meeting with the research team and all experts to finalise the CORMIX tool**

**Step 7: Develop the guidance statement for the CORMIX tool**

**Step 8: Develop an explanatory and elaboration document (E & E)**

**Fig 1. Project steps.** Moher D, Schulz KF, Simera I, Altman DG. Guidance for developers of health research reporting guidelines. PLoS Med. 2010;7(2):e1000217. PMID: 20169112; PMCID: PMC2821895.

Additionally, personal discussions with experts in the fields during relevant forums and conferences emphasized the need for an agreed-upon checklist for reporting mixed methods research to enhance its quality and benefit from a meaningful integration of qualitative and quantitative data.

**Reviewing the literature to identify potential CORMIX items.** A scoping review was conducted with the aim of extracting potential reporting items that could inform the development of a mixed methods reporting checklist.

Sources: A comprehensive search was conducted by three researchers through PubMed, EMBASE, PsycINFO, ERIC, CINAHL, ProQuest (Default selected databases: APA PsycArticles; Australia & NewZealand database; Consumer Health Database; Continental Europe Database; East & South Asia Database; East, Europe & Central Europe Database; Health & Medical Collection; Healthcare Administration Database; Library Science Database; Nursing & Allied Health Database; Psychology Database; Research Library; Social Science Database; Sociology Database; ProQuest One Academic) and SCOPUS, reviewer's guidelines of major journals (Journal of Mixed Methods Research; International Journal of Social Research Methodology; Mixed Methods Research for Nursing and the Health Sciences, International Journal of Multiple Research Approaches), and grey literature (Google Scholar and ProQuest Dissertation & Thesis Global). A complementary search using similar keywords was also conducted in Google Scholar, and the first 100 hits were screened for inclusion, in addition to screening references of relevant articles.

Keywords: The following keywords and their synonyms were used: mixed methods, multi-methods, reporting guidelines, reporting standards and reporting checklist. Boolean operators AND/OR were used. The search strategy for each database was adapted slightly due to their respective technical differences. No limits were applied. The search period covered database inception until the date of search (September 2024). The search strategy used in each database, along with the date, and number of hits is available in the supplementary file.

Inclusion and Exclusion: Articles that discuss reporting criteria or a checklist for mixed methods research published in English were included. Articles that describe how to conduct a mixed methods study or focus on quality assessment criteria were excluded.

Screening: All hits from the databases and journals were uploaded to Rayyan tool, which was used for screening the title/abstract by two researchers independently. Rayyan is an online platform that allows researchers to conduct reviews efficiently; it streamlines the process of screening and selecting relevant studies by providing a user-friendly interface for importing, organizing, and reviewing research articles [42]. Titles and abstracts were screened to include relevant articles, this was followed by full-text screening. As a supplementary search, we also reviewed selected quantitative and qualitative checklists prioritizing those frequently cited in mixed methods literature or those that have significantly influenced reporting practices. We limited our in-depth review to the top 10 most relevant checklists for each methodology (quantitative and qualitative) based on citations. Disagreements were resolved following full-text screening through discussion across the research team.

Data extraction: Items relevant to mixed methods research reporting were extracted from the included articles, using a standardised data extraction sheet developed in Excel. The data extraction sheet included the following: Article title, publication year, authors, journal, article objective, reporting items relevant to mixed methods (this includes items described in the context of mixed methods, qualitative and quantitative research). Two researchers independently extracted the data, and any discrepancies were discussed with the research team to reach a consensus.

Quality assessment: No quality assessment was conducted as the purpose of conducting the scoping review is to generate items for the Delphi phase and not to synthesize concluding evidence or recommendations.

Artificial intelligence-based large language model (paid version of ChatGPT): An AI-based Chatbot was used as a complementary tool to generate suggestions for reporting criteria of mixed methods research. The primary function of ChatGPT in this context is to provide additional perspectives and ideas that may complement the findings from the literature search. Chatbot, equipped with language processing, can develop items that can be beneficial for CORMIX. Items generated by ChatBot were compared with items found from the literature search to supplement the findings. The

intention is not to rely solely on ChatGPT, but to use it as a brainstorming tool to ensure comprehensiveness and potentially identify novel reporting elements that human researchers might overlook. Example prompts that were used "What are essential elements to report in a mixed methods study?; What information should be included when describing the integration of qualitative and quantitative data in mixed methods research." We rephrased questions and provided varied background information to elicit diverse perspectives and more comprehensive answers from the AI. We also asked the AI the same questions at various times to capture potential variations in responses due to model updates or dynamic knowledge. It is important to note that all AI-generated suggestions were critically evaluated by the research team and not automatically included in the final checklist without expert review and consensus. The supplementary file provides screen shots of examples of the chats.

We have completed the scoping review and the questionnaire development for the Delphi phase. A total of 13,439 articles were identified from the databases search. Following duplicates removal, a total of 4978 articles were screened for title and abstract by two independent researchers. Full text screening was completed for 121 articles out of which 50 articles were included. Fig 2 summarises the article retrieval process.

Items related to reporting of mixed methods were extracted by two independent researchers and reviewed to generate the preliminary list of items for the Delphi phase. Pre-liminary items are provided in the supplementary file.

## Modified Delphi phase

A modified Delphi will be conducted among the experts identified by the research team. A modified Delphi draws on relevant literature to compile an initial list of items for the expert panel to assess. This provides a strong starting point informed by evidence rather than developing items from scratch. A literature-informed initial set of items ensures important evidence-based items are not missed through reliance on expert recall alone [31].

**Experts' selection.** Expert identification will involve a multi-faceted approach to ensure comprehensive representation across disciplines. For this study, experts are defined as individuals with demonstrated knowledge and skills in mixed methods research. The selection criteria are defined in Table 1. We will begin by extracting author information from our scoping review of mixed methods literature. This will be supplemented by a targeted literature search in Scopus and Web of Science using terms such as 'mixed methods research' and 'mixed methodology' to identify authors who have published extensively in this field. We will focus on authors who have published multiple mixed methods studies or methodological papers about mixed methods research. Authors' institutional profiles and Google Scholar profiles will also be reviewed to confirm their ongoing engagement with mixed methods research. The lead author will compile an initial list of experts, categorized by research discipline, which will be reviewed and supplemented by the full research team to ensure comprehensive representation. The experts' biographies and publication records will be consulted to verify their research design expertise, including their research interests, courses taught, and scientific publications. Furthermore, we will contact the Mixed Methods International Research Association and other relevant organizations to nominate experts based on pre-determined selection criteria (Table 1). To address disciplinary differences, we will strive for a balanced representation of experts across various fields, including but not limited to psychology, sociology, health sciences, and education. This diversity will help ensure that the reporting tool captures nuances specific to different disciplines. Additionally, we will employ snowball sampling, asking identified experts to nominate peers who meet the inclusion criteria. This approach will help us access a wider network of qualified experts. Upon their agreement, an online meeting will be set up to explain the objective of the study and reduce the risk of attrition in subsequent Delphi rounds. We acknowledge several limitations in our expert identification strategy. First, reliance on academic publications may underrepresent practitioners and experts who primarily contribute through other channels. Second, English-language dominance in major databases could limit identification of experts from non-English speaking regions. Third, citation metrics and publication counts, while useful indicators, may not fully capture expertise, particularly for emerging scholars or those focusing on specific disciplinary applications of mixed methods. To mitigate these limitations, we will actively seek

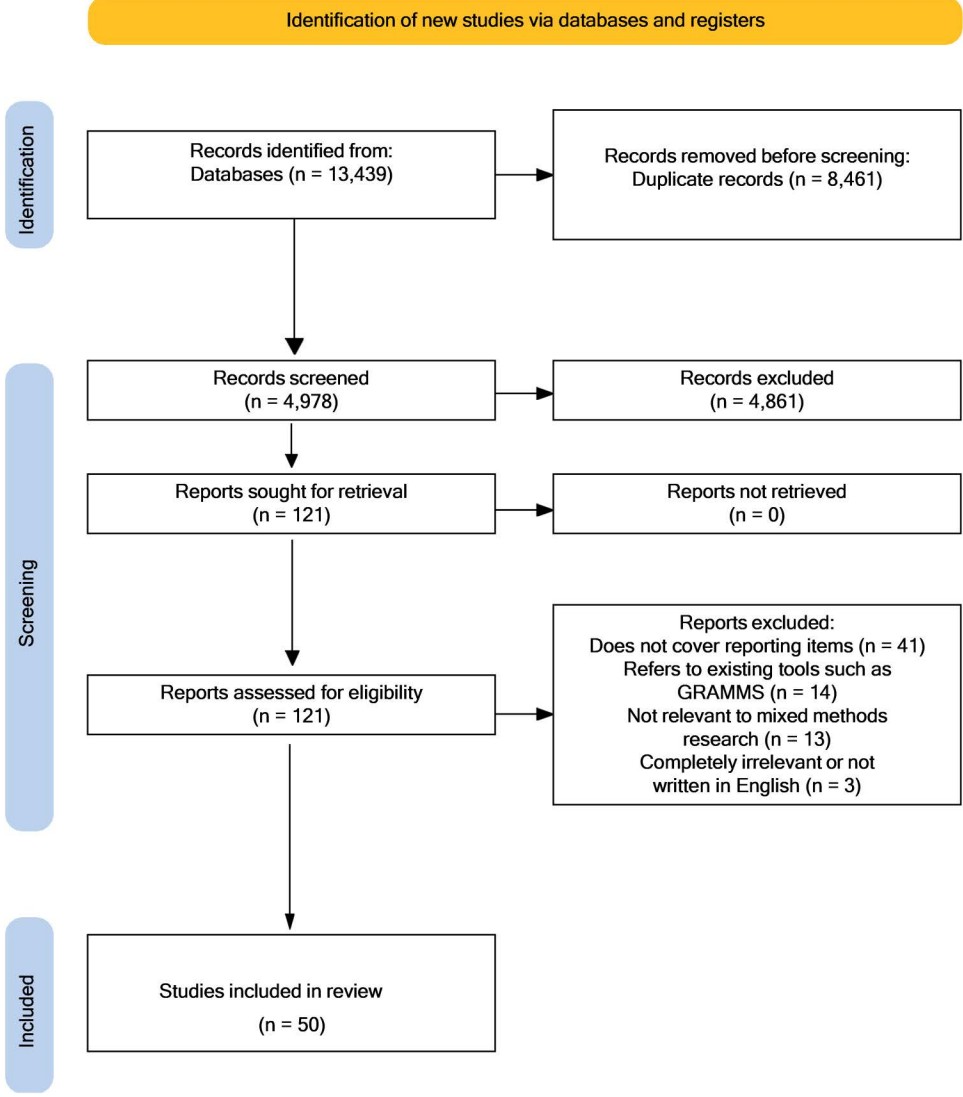

**Fig 2. Prisma flow diagram.** Haddaway, N. R., Page, M. J., Pritchard, C. C., & McGuinness, L. A. (2022). PRISMA2020: An R package and Shiny app for producing PRISMA 2020-compliant flow diagrams, with interactivity for optimised digital transparency and Open Synthesis Campbell Systematic Reviews, 18, e1230. https://doi.org/10.1002/cl2.1230Downloadcitation(.ris).

**Table 1. Criteria for Expert Selection.**

| Expert category | Qualification/Degree | Work experience |
|---|---|---|
| Journal Editors of methodology related journals such as the Journal of Mixed Methods Research | PhD or equivalent degree. | Minimum 3 years of experience as a journal editor in a journal that publishes mixed methods research |
| Content experts/ academics | Have a PhD or equivalent degree as recognised in any subject (research methodology, mixed methods research, research quality evaluation). | Minimum of 5 years of experience in conducting, publishing, or teaching mixed methods research OR at least three published mixed methods studies or significant contributions to mixed methods literature |

recommendations from professional organizations and use snowball sampling to identify experts who might be missed through traditional academic metrics. We will also make specific efforts to include experts from diverse geographical regions and academic traditions. The minimum number of experts set for this study is 40 [31,43]. Results of the Delphi will be considered compromised if the response rate drops below 75%; hence, 50 experts will be contacted to participate in this study. The recruitment period is expected at the end of November, while the Delphi phase will start in December 2024.

To ensure an optimal response rate, the following strategies will be used: (1) More than 40 experts will be contacted to participate in the study; (2) An online meeting will be set up with the experts individually before sending the questionnaire; (3) The identified experts will be given three weeks to respond to the questionnaire; (4) The deadline for questionnaire completion will be extended for one week; (5) Two reminders will be sent out to the experts on weeks three and four to complete the questionnaire; (6) If possible an online meeting will be held with the experts individually between the Delphi rounds to share the main results; (7) Strategies 4 and 5 will be repeated in the subsequent Delphi rounds. If consensus is not achieved in the second round, a third round will be conducted and strategies 4, 5, and 6 will be repeated.

**Questionnaire instrument.** Items identified from the scoping review and AI chat in the initial phase of the study were compiled into a Microsoft Word document. These items were organized into sections (items related to introduction, methodology, results and discussion). A meeting was set up with the research team to build an agreement on the number and order of items within the questionnaire. The questionnaire then underwent face and content validation were conducted by a panel consisting of the research team members and three external experts. The research team includes five members with expertise in mixed methods research, research methodology, and questionnaire design. The three external experts, identified through professional networks, have extensive experience in conducting Delphi studies. In total, eight individuals were involved in this validation process.

Each item in the questionnaire was independently evaluated by all panel members for clarity. The research team provided comments (related to content and clarity) to add/delete/change any item on a Word document. Comments were compiled and addressed. This approach ensured that every item received multiple assessments from individuals with diverse expertise. The validation process also included evaluating the comprehensibility and unambiguity of each item, identifying any potential gaps in the questionnaire; estimating the time required to complete the questionnaire, and assessing the cognitive load on the participant. This was followed by a face-to-face meeting to finalise the item list and the questionnaire. The questionnaire items are available in the supplementary file. The next step will be to upload the questionnaire into Google Forms and share it with the experts to be completed anonymously. The online question-naire will include the following: (1) A brief introduction to the study, including the objectives; (2) Instructions on how to complete the questionnaire and the expected timeframe; (3) Demographics section of the experts; (4) Pre-set CORMIX items for rating on a scale of 1–10, where 1 represents strongly disagreeing with keeping the item and 10 represents strongly agreeing with keeping the item within the CORMIX. Each item will be followed by a free text box to allow the expert to insert any comments they have. (5) A section for recommending other items not originally included with justification.

**Defining consensus.** A 10-point Likert scale will be used for experts to rate each item on suitability to be included in the CORMIX, i.e., how much they agree to retain the item in the final checklist. Consensus for this study is defined by the sum percentage of scores 7–10 on the Likert scale representing "agree to keep the item" ratings by ≥ 85% of the experts in the first round. The 85% threshold for consensus is based on commonly used cutoffs in Delphi studies for reporting checklist development [29,31]. Thus: a. ≥ 85% high consensus; b. 80%–84% moderate consensus; c. 75%–79% low consensus; d. ≤ 74% poor or no consensus. Based on the literature, it is assumed that two rounds of modified Delphi will be conducted among experts to reach a consensus [29,31]. Subsequent rounds will utilize a 75% cut-off for consensus.

Experts will also be asked to rate each item for clarity on a 3-point Likert scale (1- completely unclear, 2- somewhat clear, and 3 – completely clear). They will also have an open text following each item to provide any suggestions for modifying the item and to justify their rating of the item.

**Second round of modified Delphi.** Following the completion of the first round of the Delphi, the results and comments will be compiled, and adjustments to items will be made. Assuming that consensus on all items is yet to be reached, the second round of the Delphi will follow the same steps as the first round; however, upon sharing the questionnaire with experts, an additional document will be shared, which will include the collective comments and results of the first round as well as a summary of the changes made to the items. Participants will be given the opportunity to re-rate the items taking into consideration the collective comments from round one.

**Third round of modified Delphi.** We anticipate reaching a consensus with two Delphi rounds based on similar research [34]. Nonetheless, if consensus is not achieved with two rounds, a third round will be conducted and will follow the same steps as the first and second rounds.

**Delphi data analysis.** Following each round of Delphi, data will be analysed and shared with the experts. Analysis of quantitative data will be performed using Statistical Package for Social Science (SPSS version 29). Descriptive statistics (median and interquartile range) will be used for Likert scale items (i.e., ordinal data). Qualitative data (i.e., comments made by experts) will be compiled for adjusting the subsequent Delphi rounds and presented in a table format.

## Finalizing the CORMIX phase

Following the Delphi phase, a meeting will be set up as face-to-face for the research team while Delphi panel experts will be asked to join virtually to go over the CORMIX items for finalization. In this meeting, the final items will be shared, along with the analysis process. Upon confirmation, the checklist will be piloted by a team of two PhD students and eight mixed methods researchers. The PhD students will be recruited from our institution's research methodology program, while the eight researchers will be identified through a combination of purposive sampling from our professional networks and open calls in relevant research forums. All piloting members will have extensive training and expertise in conducting and publishing mixed methods studies. This piloting team will independently apply the checklist to the same set of 15 sampled mixed methods studies across different fields. These studies will be selected through convenience sampling to include studies from various disciplines. The fifteen articles will be identified through a search in Scopus and PsycINFO for mixed method studies published in the last three years. To ensure balanced representation, articles will be stratified by major disciplines (psychology, sociology, health sciences, and education), aiming for approximate balance across fields where feasible. This disciplinary stratification will help ensure the checklist's applicability across different fields. Feedback will be collected quantitatively using a structured questionnaire used after applying the checklist to each study. The questionnaire will include rating item clarity, relevance, and ease of use on a five-point Likert scale. Qualitative comments will also be collected using open-ended format questions where the rater will document specific issues, limitations, or suggestions for each checklist item and the overall tool. The pilot team will then convene to consolidate their feedback and any recommended modifications to the checklist. This input will be used in revising and finalizing the checklist.

## Guidance statement

To accompany the final checklist, the research team will develop a guidance statement. The 2–3 page guidance statement will provide background context, rationale for the checklist development process, an overview of the checklist domains, items, and how to apply them including instructions, question prompts, examples, and tools and resources to facilitate appropriate use. The guidance statement will undergo pilot testing and feedback from intended end-users such as journal editors, study authors, and peer reviewers selected through an open invite to Delphi experts and personal contact, and individuals who participated in the piloting of the CORMIX tool. The piloting will be conducted similar to the piloting of the CORMIX where five end-users will review and pilot the guidance statement and answer a pre-set questionnaire asking about clarity, comprehensiveness and usability of the guide. Feedback from these reviewers will be carefully considered and incorporated into the final version of the guidance statement. This process aims to ensure that the final product is user-friendly, comprehensive, and applicable across different contexts in mixed methods research reporting.

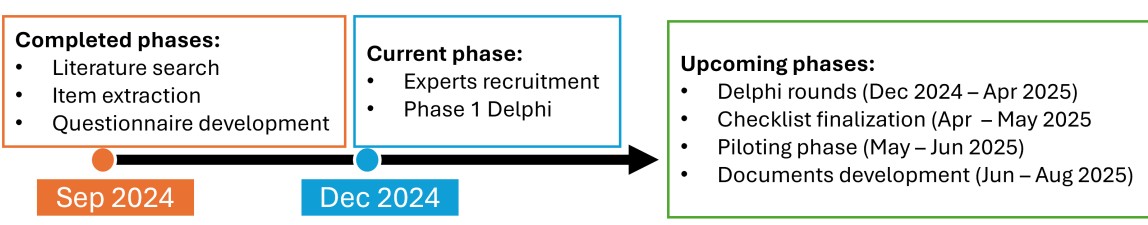

**Fig 3. Completed steps and timeline.**

### Developing an explanation and elaboration document (E&E)

A detailed justification document will be developed by the research team explaining the rationale and details of each item within the CORMIX tool and published in a peer-reviewed journal. The justification document will outline the methodological underpinnings, examples from the literature, expert perspectives, and consensus data underpinning each item's inclusion in the final checklist. This detailed justification document will provide transparency on the deliberative process involved in creating the checklist. It will also facilitate the peer review process when publishing the CORMIX tool by allowing reviewers to clearly evaluate the evidentiary basis for the checklist criteria.

### Completed steps

We have completed the initial phase of literature review and the questionnaire development for the Delphi phase. We are currently recruiting experts and running the first phase of the Delphi process. Fig 3 below highlights the completed steps.

### Discussion

The Consolidated Checklist for Reporting Mixed Methods Research (CORMIX) aims to provide clear, evidence-based guidance to enhance the quality and transparency of mixed methods studies. CORMIX will encompass a reporting checklist for mixed methods research, a user guide for applying the checklist, and a justification document outlining the rationale for each checklist item. Through a rigorous development process informed by EQUATOR Network guidelines, we will engage experts to build consensus on essential criteria that should be reported in mixed methods studies across all phases. The overarching goal is for CORMIX to serve as a valuable framework that guides researchers in conducting and reporting quality mixed methods research.

### Supporting information

**S1 File. Supporting information.**
(DOCX)

### Acknowledgments

The publication of this article was funded by Qatar National Library.

### Author contributions

**Conceptualization:** Myriam Jaam, Ahmed Awaisu, Derek Stewart, Banan Mukhalalati, Muhammad Abdul Hadi.

**Data curation:** Myriam Jaam.

**Formal analysis:** Myriam Jaam, Ahmed Awaisu, Derek Stewart, Banan Mukhalalati, Ahsan Sethi, Marwa Elshazly, Abrar Abdelrahman, Muhammad Abdul Hadi.

**Investigation:** Myriam Jaam, Ahmed Awaisu, Derek Stewart, Banan Mukhalalati, Marwa Elshazly, Abrar Abdelrahman, Muhammad Abdul Hadi.

**Methodology:** Myriam Jaam, Derek Stewart, Banan Mukhalalati, Ahsan Sethi, Marwa Elshazly, Abrar Abdelrahman, Muhammad Abdul Hadi.

**Project administration:** Myriam Jaam, Ahmed Awaisu, Muhammad Abdul Hadi.

**Resources:** Muhammad Abdul Hadi.

**Software:** Muhammad Abdul Hadi.

**Supervision:** Ahmed Awaisu, Muhammad Abdul Hadi.

**Validation:** Myriam Jaam, Ahmed Awaisu, Derek Stewart, Banan Mukhalalati, Ahsan Sethi, Muhammad Abdul Hadi.

**Visualization:** Myriam Jaam, Muhammad Abdul Hadi.

**Writing – original draft:** Myriam Jaam, Ahmed Awaisu, Derek Stewart, Banan Mukhalalati, Ahsan Sethi, Muhammad Abdul Hadi.

**Writing – review & editing:** Myriam Jaam, Ahmed Awaisu, Derek Stewart, Banan Mukhalalati, Ahsan Sethi, Marwa Elshazly, Abrar Abdelrahman, Muhammad Abdul Hadi.

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
