## [Decision Letter · Decision Letter 0]

23 Jul 2024

PONE-D-24-20917Protocol for Developing a COnsolidated Checklist for Reporting MIXed-Methods Research (CORMIX) Using Modified DelphiPLOS ONE

Dear Dr. Jaam,

Thank you for submitting your manuscript to PLOS ONE. After careful consideration, we feel that it has merit but does not fully meet PLOS ONE’s publication criteria as it currently stands. Therefore, we invite you to submit a revised version of the manuscript that addresses the points raised during the review process.

We look forward to receiving your revised manuscript.

Kind regards,

Sergi Fàbregues

Academic Editor

PLOS ONE

2. Please provide additional details regarding participant consent. In the ethics statement in the Methods and online submission information, please ensure that you have specified (1) whether consent will be informed and (2) what type you obtain (for instance, written or verbal, and if verbal, how it will be documented and witnessed).

Editor Comments:

- On page 4, lines 62-66, the authors imply that mixed methods research (MMR) leads to a better understanding of the phenomenon under study and increase the quality of the results. Conversely, mixed methods do not increase the quality of a study but rather allow researchers to answer complex questions, among other things.

- Throughout the paper, the authors use the term triangulation as a synonym for integration. The MMR literature describes triangulation as one of the possible rationales for MMR; therefore, using the term to describe integration is inaccurate.

- The authors do not cite the journal article reporting criteria for MMR developed by the American Psychological Association (Levitt et al., 2018). This omission is concerning because these standards fill two important gaps identified by the authors: (1) they were developed by a task force group assembled specifically for this purpose and (2) they provide a detailed and comprehensive set of reporting guidelines for MMR.

- The Methods section needs to be much more detailed, especially the literature review section, which is weak in terms of reporting. The authors must specify the exact search query that they will use. Additionally, what are the inclusion and exclusion criteria?

- It is unclear why the terms quantitative and qualitative will be used as separate terms in the search (instead of: (quantitative AND qualitative)), and why articles discussing checklists for quantitative and qualitative research will also be included in the review. There are a large number of checklists in both methodologies and reviewing them all will certainly involve a lot of unnecessary work. Moreover, why review these checklists if the goal is to develop reporting guidelines for MMR?

- As noted by one of the reviewers, the authors seem to confuse reporting quality with methodological quality. For instance, the MMAT is a tool to assess the methodological quality of MMR studies, not the reporting quality, as the authors currently imply.

- On page 8 the authors describe the weaknesses of existing reporting tools. Specifically, they mention that they are based on the views of the authors, have limited guidance for use, and do not use a robust methodology. These statements should be substantiated. For example, the MMAT is based on a Delphi study with experts (not the views of the authors) and has a detailed manual describing how to use the tool (therefore, it does not have limited guidance for use).

- As one of the reviewers argues, what is the function of ChatGPT?

- The sampling strategy has several important weaknesses. Selecting the sample from personal contacts and snowball sampling may miss MMR researchers with important expertise on the topic. Instead, a strategy to identify researchers based on a literature search would be much more appropriate and accurate. Furthermore, why should content experts have a rank of associate professor or higher? How will consumer experience with mixed methods be determined? In addition, we know from the literature that researchers conceptualize and operationalize MMR quality differently across disciplines. Furthermore, each discipline has different reporting standards. How will the issue of disciplinary differences be addressed in the sample?

- In conclusion, this protocol has a number of important issues that need to be addressed by the authors. Acceptance of the paper will depend on the authors' ability to address these issues and to make substantial changes in some parts of the study design.

Reviewers' comments:

Reviewer's Responses to Questions

**Comments to the Author**

1. Does the manuscript provide a valid rationale for the proposed study, with clearly identified and justified research questions?

Reviewer #1: Yes

Reviewer #2: Partly

Reviewer #3: Partly

2. Is the protocol technically sound and planned in a manner that will lead to a meaningful outcome and allow testing the stated hypotheses?

Reviewer #1: Yes

Reviewer #2: Partly

Reviewer #3: No

3. Is the methodology feasible and described in sufficient detail to allow the work to be replicable?

Reviewer #1: Yes

Reviewer #2: No

Reviewer #3: No

4. Have the authors described where all data underlying the findings will be made available when the study is complete?

Reviewer #1: No

Reviewer #2: No

Reviewer #3: Yes

5. Is the manuscript presented in an intelligible fashion and written in standard English?

Reviewer #1: Yes

Reviewer #2: Yes

Reviewer #3: Yes

6. Review Comments to the Author

You may also provide optional suggestions and comments to authors that they might find helpful in planning their study.

Reviewer #1: I am of two minds about this article. First, I am not a supporter of checklists in non-quantitative research. Second, if one were to develop such a checklist for mixed methods research, the Delphi procedure that the authors propose would be well-suited to that task.

The origin of my doubts with checklists comes from their use in qualitative research. In particular, the authors refer to SRQR and COREQ, without considering the controversy surrounding such “check the box” ratings of qualitative research (for an early critique, see Barbour, 2001, which has over 4,000 citations in Google Scholar).

The underlying problem is that qualitative methods are so diverse that no one set of criteria can apply to even a majority of the most widely used approaches (e. g., grounded theory, interpretive phenomenological analysis, reflexive thematic analysis, and qualitative content analysis). The broad range of criteria that these methods use to assess rigor is quite different from the far more limited criteria for quantitative methods such as surveys and experiments.

This debate around the use of checklists in qualitative methods poses at least two challenges for these authors: Will they limit their Delphi participants to researchers who agree with their preference for universal standards? And, will they try to apply the same standards to both the qualitative and quantitative methods in their instrument?

Moving on to issues related to the authors’ proposed methods, I am quite familiar with Delphi studies, and I found the design and description of the current study to be very thorough, with one small exception. On p. 11, lines 222 -223 there is a reference to having the second round of the expert panel review comments from the first round, but there was no mention of collecting such comments in the first round. I assume this just an omission in describing the first round, since it is traditional in a Delphi to collect such comments along with the numeric ratings.

In contrast, I was less satisfied with the section on Finalizing the instrument (p. 13), which relies on a sample of five articles from “different fields.” First, five seems like a very low number, given that the authors expect their checklist to have widespread utilization. Second, I would prefer to see some systematic procedure for selecting those articles.

Overall, the nature of this article as a protocol leads me to emphasize the strengths of the proposed methods for meeting the authors’ stated goals. Still, I think the authors should be more reflexive in their thinking about whether a set of check-the-box criteria are indeed adequate for such a complex task as assessing reporting quality in mixed methods research.

Reference

Barbour, R.S. (2001). Check lists for improving rigour in qualitative research: A case of the tail wagging the dog? BMJ, 232:1115-1117.

Reviewer #2: General Comments/Summary of the paper

This manuscript presents a protocol for the development of a reporting guideline for mixed methods research. The guideline that will be developed from this project can be relevant for those interested in mixed methods research. The main comments concern the confusion between reporting and methodological quality and the lack of information in the methods section. The comments are detailed below.

Main comments

1. The authors use the terms “mixed methods”, “mixed-method” and “mixed-methods” interchangeably throughout the manuscript (text, table and figure). Textbooks usually use “mixed methods” without the hyphen and with an S at the end of method.

2. There is a confusion between reporting and methodological quality tools in the introduction (page 5) and initial phase (page 8). For example, they cited the MMAT, QATSDD and QuADS as existing tools that they found (line 137). Yet, these tools were not developed as reporting guideline but for appraising the methodological quality of studies. Reporting guidelines will usually include more criteria to ensure that the information is complete and clear. There are more than 600 reporting guidelines on the EQUATOR Network. More than a dozen mixed methods reporting tools can be found from the EQUATOR Network such as the MMR-RHS checklist and ASSESS tool. The authors should focus on these tools to justify their project and develop their questionnaire. There are also other reporting guidances for mixed methods that the authors can rely on such as Brown, K., Elliott, S., Leatherdale, S., & Robertson-Wilson, J. (2015). Searching for rigour in the reporting of mixed methods population health research: a methodological review. Health Education Research, 30(6), 811-839; Leech, N., & Onwuegbuzie, A. (2010). Guidelines for conducting and reporting mixed research in the field of counseling and beyond. Journal of Counseling and Development, 88(1), 61-70; Burrows, Timothy J. 2013. A preliminary rubric design to evaluate mixed methods research. Ph.D. diss., Virginia Polytechnic Institute and State University, https://www.proquest.com/dissertations-theses/preliminary-rubric-design-evaluate-mixed-methods/docview/1513244026/se-2. The authors should consider reviewing this sentence “…these tools use different sets of criteria based on author’s views and experts opinions rather than robust methodology. Standardized scoring system is lacking, and subjective judgement is made by accessors, hence, inter-rater reliability may be inconsistent. (lines 141-143)”. Some tools have been developed using a rigorous process (including Delphi and qualitative interviews). Also, inter-rater reliability is not necessarily a recommended step for the development of reporting guideline (Moher et al., 2010).

3. More information on the research team is needed (page 8). How many persons are part of the team? Are they junior/senior researchers or trainees? What is the expertise of each member? Do they all have expertise/experience in mixed methods? Based on the reference list, it seems that only the last author has expertise in mixed methods.

4. In figure 1 (page 7), there is a step (“Step 6: Conduct a face-to-face meeting with the research team and all experts to finalise the CORMIX tool”) that is not described in the text. More information is needed on how this step will be conducted. Who are “all the experts” (e.g., those that participated in the Delphi?)?

5. Initial phase (page 9) : 1) 7 databases are listed on line 152. Yet, ProQuest is a platform that provides access to several databases. The authors should indicate which databases from ProQuest have been used; 2) The search strategy is not detailed. Did they consulted a specialized librarian to develop the search strategy? Did they used Boolean, truncation or proximity operators? What time frame was covered in the search strategy?; 3) The authors mentioned searching in the grey literature (line 153). Is the search of the grey literature limited to Google Scholar? If no, please add the other sources used. Also, the “major journals” searched should be listed (line 153); 4) The authors mentioned that the data will be extracted using a “Standardised data extraction sheet” (line 161). More information on what data will be extracted is needed; 5) The authors mentioned, “The use of AI to generate item Chatbot, equipped with language processing, can develop items that can be beneficial for CORMIX. Items generated by ChatBot will be compared with items found from the literature search to supplement the findings. To ensure comprehensiveness, the prompt questions will be repeated in different timestamps and contexts.”(lines 164-167). This method is innovative but would benefit from more precision. For example, what will be asked to generate the items? Is this process replicable by other researchers (e.g., if another researcher use ChatBot with the same questions, will the same items be generated?). Also, what is meant by “repeated in different timestamps and contexts”?

6. Modified Delphi phase (page 10): 1) More information on how the experts will be identified and recruited is needed; 2) The authors mentioned aiming for a minimum of 20 experts. What is the rationale behind this number? Is this number after the 1st round or after three rounds? They mentioned that more than 20 (line 181). How many more?; 3) The recruitment is expected to start in July 2024 and data collection in August 2024. Thus, the questionnaire and phase 1 is completed. The authors could provide more detailed information on the Initial phase and questionnaire development; 4) The authors mentioned that “(6) An online meeting will be held with the experts between the Delphi rounds to share the main results” (lines 188-189). This is not a typical step in a Delphi method. The results of a round are usually shared at the beginning of the questionnaire of the next round. Could this online meeting influence anonymity (a characteristic of a Delphi method)?; 5) It is mentioned that the research team will perform a face and content validation of the questionnaire (line 197). How many persons will rate each item? What are their expertises?

7. Finalizing the CORMIX Phase (page 13): How will the raters (2 PhD students and 5 mixed methods researchers) be recruited? How will the 5 published mixed methods studies identified? Will they choose good and bad reported papers? How will the issues or limitations identified from the raters be collected (e.g., using a pre-developed questionnaire vs during an interview or focus group)?

8. Guidance statement: It is mentioned that the guidance statement “will undergo pilot testing and feedback from intended end-users” (lines 248-249). More information on how this testing is needed (how will they be recruited, contacted, how many, how data will be collected, …).

9. Explanation and Elaboration (E&E) Document: Will the E&E also be submitted to testing and feedback as the guidance statement?

10. The authors should be careful on their use of the word “triangulation” (lines 66, 76, 147). Triangulation refers of the use of multiple methods or sources of data. In the context of their sentences, they seem to refer more to integration than triangulation. For example, at this sentence, “These reported limitations highlight that researchers merely use mixed methods as a means to collect large volume of qualitative and quantitative data with limited triangulation.”, the word triangulation could be replaced by “integration”.

Minor comments

Line 80: write Equator with capital letters.

Line 112: replace EQUATER by EQUATOR

Line 118: remove “1.” at the end of the sentence.

Line 281: Reference #5 is incomplete.

Reviewer #3: More information is needed about the initial phase to clarify the process for the reader, including what the Rayyan tool is, how using ChatGPT benefits the project at this phase, and what you will do with the ChatGPT results.

Expert selection: The expert selection process is a bit confusing. First, just because someone is a journal editor of a journal that publishes mixed methods research does not mean that they are an expert in mixed methods quality (unless you are only recruiting from methodological journals such as JMMR, but that does not appear to be the case). Additionally, requiring that content experts/academics be at the rank of associate professor or higher may be limiting. In some fields, mixed methods research is relatively new and assistant professors may know more about it than full professors. For the consumers, how will you know they have a minimum of 3 years of experience in using and publishing mixed-method research?

7. PLOS authors have the option to publish the peer review history of their article (what does this mean? ). If published, this will include your full peer review and any attached files.

**Do you want your identity to be public for this peer review?** For information about this choice, including consent withdrawal, please see our Privacy Policy .

Reviewer #1: No

Reviewer #2: No

Reviewer #3: No

---

## [Author Response · Author response to Decision Letter 1]

3 Sep 2024

Sep 2rd, 2024

The Editor-in-Chief

PLOS ONE

Dear Dr. Fàbregues

Revision of Manuscript

PONE-D-24-20917

Protocol for Developing a COnsolidated Checklist for Reporting MIXed-Methods Research (CORMIX) Using Modified Delphi.

The authors of the above-named manuscript would like to thank you for your e-mail dated 24 July 2024, containing the constructive comments for the manuscript. The authors very much appreciate the comments and have revised the manuscript accordingly as attached. We are positive that the comments helped in enhancing the quality and scientific merit of the paper.

Below are separate pages, detailing our responses to the comments of the journal requirements and reviewer.

Please note that changes are made using “track changes” tool within the revised manuscript.

Thank you to the editor and the reviewer for the time spent on this manuscript. We look forward to receiving your kind response.

Sincerely yours,

The Corresponding Author

All in all, we thank you for your great review which we believe would add great value to the adjusted manuscript.

Detailed Responses to the Comments of the Journal Requirements, Editor and Reviewers

We have addressed the formatting requirements as per the documents provided.

2. Please provide additional details regarding participant consent. In the ethics statement in the Methods and online submission information, please ensure that you have specified (1) whether consent will be informed and (2) what type you obtain (for instance, written or verbal, and if verbal, how it will be documented and witnessed).

We added “Informed written consent will be obtained from experts participating in the Delphi phase through email prior to completing the questionnaire” within the ethical approval section.

This manuscript presents only as a protocol for the study. No primary data is available at this stage.

Editor Comments:

- On page 4, lines 62-66, the authors imply that mixed methods research (MMR) leads to a better understanding of the phenomenon under study and increase the quality of the results. Conversely, mixed methods do not increase the quality of a study but rather allow researchers to answer complex questions, among other things.

We thank the editor for this insightful observation. Upon reflection, we concur with the assessment regarding mixed methods research (MMR). We removed the reference to increasing "the quality of the study" and instead emphasized MMR's capacity to address complex research questions. This modification aligns more closely with the current understanding of MMR's role in research design and methodology.

- Throughout the paper, the authors use the term triangulation as a synonym for integration. The MMR literature describes triangulation as one of the possible rationales for MMR; therefore, using the term to describe integration is inaccurate.

We appreciate the editor's observation regarding our use of terminology. Upon careful review, we acknowledge that our usage of 'triangulation' as a synonym for 'integration' throughout the paper is indeed imprecise. In light of this we revised these terms throughout the paper.

- The authors do not cite the journal article reporting criteria for MMR developed by the American Psychological Association (Levitt et al., 2018). This omission is concerning because these standards fill two important gaps identified by the authors: (1) they were developed by a task force group assembled specifically for this purpose and (2) they provide a detailed and comprehensive set of reporting guidelines for MMR.

We sincerely appreciate the editor bringing this significant oversight to our attention. We acknowledge the importance of Levitt et al.'s (2018) work on Mixed Methods Article Reporting Standards (MMARS) developed by the American Psychological Association task force. We have rectified this omission by incorporating a discussion of MMARS in our manuscript as follows: “The MMARS provides guidelines for reporting mixed methods research in psychology and related fields. While more comprehensive than previous tools, it does not provide the level of detail some researchers might need for comprehensive reporting across all aspects of a mixed methods study. Furthermore, its reliance on cross-referencing with other reporting tools (JARS-Qual and JARS-Quant) can complicate the reporting process, potentially creating challenges for researchers attempting to adhere to a unified set of guidelines. This structure, while comprehensive in theory, may introduce practical difficulties in application, particularly for researchers seeking a more streamlined, self-contained reporting standard for mixed methods research.”

- The Methods section needs to be much more detailed, especially the literature review section, which is weak in terms of reporting. The authors must specify the exact search query that they will use. Additionally, what are the inclusion and exclusion criteria?

We appreciate the editor's feedback regarding the need for more detail in our Methods section. We have taken steps to address these concerns and strengthen our methodology reporting. For the literature review: We acknowledge that our initial description of the literature review process lacked details . To address this, we have expanded this section to include:

a) Search Query: We have added an example search query to illustrate our search strategy. The specific query is (("mixed method*" OR "multi-method") AND (reporting OR checklist OR guideline*)) OR ((quantitative AND qualitative) AND (reporting OR checklist OR guideline*))

b) Inclusion and exclusion criteria: We have explicitly stated our inclusion and exclusion criteria as follows: Articles which discuss reporting criteria or a checklist for mixed- methods research, quantitative, or qualitative research and published in English will be included. Articles that describe how to conduct a mixed methods study or focus on quality assessment criteria will be excluded.

- It is unclear why the terms quantitative and qualitative will be used as separate terms in the search (instead of: (quantitative AND qualitative)), and why articles discussing checklists for quantitative and qualitative research will also be included in the review. There are a large number of checklists in both methodologies and reviewing them all will certainly involve a lot of unnecessary work. Moreover, why review these checklists if the goal is to develop reporting guidelines for MMR?

We recognize the editor's concern about the potential volume of checklists. However, we believe that mixed methods research (MMR) is built upon the integration of quantitative and qualitative approaches, thus by reviewing checklists for these individual methodologies, we gain a comprehensive understanding of the reporting standards that form the foundation of MMR. Insights. By understanding the reporting requirements for individual methodologies, we can ensure that our MMR guidelines are consistent with and complementary to existing standards. To address the issue of overwork, we will implement a two-stage screening process: (1) Initial screening will focus on mixed methods-specific guidelines. (2) Quantitative and qualitative checklists will be reviewed selectively, prioritizing those frequently cited in mixed methods literature or those that have significantly influenced reporting practices.

- As noted by one of the reviewers, the authors seem to confuse reporting quality with methodological quality. For instance, the MMAT is a tool to assess the methodological quality of MMR studies, not the reporting quality, as the authors currently imply.

We sincerely appreciate the editor's observation and the reviewer's insight regarding the distinction between reporting quality and methodological quality. We acknowledge that our manuscript mixed these two concepts in some instances, particularly in our discussion of the Mixed Methods Appraisal Tool (MMAT). To address this important point, we have conducted a thorough review of our manuscript to ensure clear differentiation between reporting quality and methodological quality throughout.

Specifically, we have modified our reference to the MMAT as a quality assessment tool rather than a reporting guideline. For example, we now state: the ‘MMAT, QATSDD, and QuADS primarily emphasize quality assessment rather than comprehensive reporting guidance.'

We have clarified our focus on reporting quality and guidelines, emphasizing that while methodological quality is crucial, our current work specifically addresses the standards and practices for comprehensive reporting of mixed methods research.

- On page 8 the authors describe the weaknesses of existing reporting tools. Specifically, they mention that they are based on the views of the authors, have limited guidance for use, and do not use a robust methodology. These statements should be substantiated. For example, the MMAT is based on a Delphi study with experts (not the views of the authors) and has a detailed manual describing how to use the tool (therefore, it does not have limited guidance for use).

We appreciate the editor's comment regarding our characterization of existing reporting tools. Upon careful review, we acknowledge that our statements were overly broad and did not accurately represent the nuances of all existing tools, particularly the Mixed Methods Appraisal Tool (MMAT).

To address this, we have thoroughly revised the paragraph in question to more accurately reflect the current state of mixed methods research tools, distinguishing between reporting guidelines and quality assessment tools. We also removed generalizations about the weaknesses of existing tools and instead provide a more nuanced discussion of the gaps in the literature specifically related to reporting guidelines for mixed methods research. Additionally, we have clarified that our focus is on developing comprehensive reporting guidelines, rather than quality assessment tools. This distinction is now explicitly stated to avoid any confusion.

- As one of the reviewers argues, what is the function of ChatGPT?

The primary function of ChatGPT in this context is to provide additional perspectives and ideas that may complement the findings from the literature search. The intention is not to rely solely on ChatGPT, but to use it as a brainstorming tool to ensure comprehensiveness and potentially identify novel reporting elements that human researchers might overlook. We reviewed the paragraph to clarify the role of ChatGPT.

- The sampling strategy has several important weaknesses. Selecting the sample from personal contacts and snowball sampling may miss MMR researchers with important expertise on the topic. Instead, a strategy to identify researchers based on a literature search would be much more appropriate and accurate. Furthermore, why should content experts have a rank of associate professor or higher? How will consumer experience with mixed methods be determined? In addition, we know from the literature that researchers conceptualize and operationalize MMR quality differently across disciplines. Furthermore, each discipline has different reporting standards. How will the issue of disciplinary differences be addressed in the sample?

We appreciate the editor's and reviewer’s valuable feedback and have thoroughly revised our approach to ensure a more robust and comprehensive study. Our aim is to produce high-quality work that represents the diverse landscape of mixed methods research. As such, we have implemented the following changes:

Sampling Strategy: We will now use a multi-faceted approach to identify experts:

- Extracting author information from our scoping review

- Conducting a targeted literature search for mixed methods experts

- Contacting mixed methods organizations and prominent journals for expert nominations

- Utilizing snowball sampling as a supplementary method to reach additional experts

Inclusion Criteria: We have removed academic ranking requirements to ensure a more inclusive selection based on expertise rather than hierarchical position. Our revised criteria for experts include:

- Holding a PhD or equivalent degree in any relevant field

- A minimum of 5 years of experience in conducting, publishing, or teaching mixed methods research

- At least three published mixed methods studies or significant contributions to mixed methods literature

Disciplinary Diversity: To address the concern about disciplinary differences, we will:

- Aim for a balanced representation of experts across various fields (e.g., psychology, sociology, health sciences, education)

- During the consensus process, we will explicitly address and document how disciplinary differences in reporting standards are reconciled

Handling Disciplinary Conflicts: In the consensus-building phase, we will:

- Encourage experts to articulate discipline-specific concerns

- Use a modified Delphi technique to address and resolve conflicting viewpoints

- Aim for a set of core reporting standards applicable across disciplines, with the possibility of discipline-specific modules if necessary

Future-Proofing: As recommended by Moher et al., we commit to:

- Regularly reviewing and updating the tool to reflect methodological advancements

- Establishing a mechanism for ongoing feedback from the research community

- Conducting periodic reviews to ensure the tool remains relevant across disciplines

- In conclusion, this protocol has a number of important issues that need to be addressed by the authors. Acceptance of the paper will depend on the authors' ability to address these issues and to make substantial changes in some parts of the study design.

We thank the reviewer for their valuable insight and comments. We hope we have addressed the comments sufficiently and improved the paper clarity.

Reviewers' comments:

Reviewer #1: I am of two minds about this article. First, I am not a supporter of checklists in non-quantitative research. Second, if one were to develop such a checklist for mixed methods research, the Delphi procedure that the authors propose would be well-suited to that task.

The origin of my doubts with checklists comes from their use in qualitative research. In particular, the authors refer to SRQR and COREQ, without considering the controversy surrounding such “check the box” ratings of qualitative research (for an early critique, see Barbour, 2001, which has over 4,000 citations in Google Scholar).

The underlying problem is that qualitative methods are so diverse that no one set of criteria can apply to even a majority of the most widely used approaches (e. g., grounded theory, interpretive phenomenological analysis, reflexive thematic analysis, and qualitative content analysis). The broad range of criteria that these methods use to assess rigor is quite different from the far mor

---

## [Decision Letter · Decision Letter 1]

17 Oct 2024

PONE-D-24-20917R1Protocol for Developing a COnsolidated Checklist for Reporting MIXed-Methods Research (CORMIX) Using Modified DelphiPLOS ONE

Dear Dr. Jaam,

Thank you for submitting your manuscript to PLOS ONE. After careful consideration, we feel that it has merit but does not fully meet PLOS ONE’s publication criteria as it currently stands. Therefore, we invite you to submit a revised version of the manuscript that addresses the points raised during the review process.

The authors have addressed most of the issues raised in the first round of reviews. However, several important points still need to be clarified. Reviewer 2 highlights a number of issues in the reporting of the first phase, while reviewer 1 notes others specifically related to the Delphi phase. 

We look forward to receiving your revised manuscript.

Kind regards,

Sergi Fàbregues

Academic Editor

PLOS ONE

Additional Editor Comments:

- Provide a rationale for the feasibility of the review and questionnaire development according to the stated timetable and clarify whether these stages have been completed.

- On line 94 you say: “Attempts have been made to formulate reporting guidelines for mixed methods research to improve quality, but comprehensive standards are still lacking”. What do you mean by comprehensive? Please explain further why the published reporting frameworks are not comprehensive.

- On line 106, when discussing the JARS, you state: “While more comprehensive than GRAMMAS, it does not provide the level of detail some researchers might need for comprehensive reporting across all aspects of a mixed methods study”. Please explain this statement further, as we believe that the JARS is quite detailed.

- On line 124, you state that the tool will address “the philosophical underpinnings of mixed methods”. What are these underpinnings? How will they be addressed?

- Include the full search strategy used or planned as a supplementary file.

- Clarify the issues raised by reviewer 2 regarding the screening process and the selection of checklists for review.

- If the review has been completed, please indicate the implications of not following the reviewers’ suggestions and how these issues will be addressed. We believe that the reviewers’ comments are important enough to be considered.

- Provide more details on how mixed methods experts will be identified, both in terms of the literature to be searched and the criteria for selecting experts.

- Consider reviewer 1’s suggestions regarding measurement tools, criteria for agreement, and other methodological issues.

- Clarify how the 10 articles for the pilot will be selected.

- Respond to any other issues raised by the reviewers.

- There are several typographical and grammatical errors throughout the paper. These need to be corrected: e.g. lines 64-65, 101-102.

Reviewers' comments:

Reviewer's Responses to Questions

**Comments to the Author**

1. Does the manuscript provide a valid rationale for the proposed study, with clearly identified and justified research questions?

Reviewer #1: Yes

Reviewer #2: Yes

2. Is the protocol technically sound and planned in a manner that will lead to a meaningful outcome and allow testing the stated hypotheses?

Reviewer #1: Yes

Reviewer #2: Partly

3. Is the methodology feasible and described in sufficient detail to allow the work to be replicable?

Reviewer #1: Yes

Reviewer #2: Yes

4. Have the authors described where all data underlying the findings will be made available when the study is complete?

Reviewer #1: Yes

Reviewer #2: No

5. Is the manuscript presented in an intelligible fashion and written in standard English?

Reviewer #1: Yes

Reviewer #2: Yes

6. Review Comments to the Author

You may also provide optional suggestions and comments to authors that they might find helpful in planning their study.

Reviewer #1: A this point, I still have a number of comments, most of which are rather small.

On lines 117-118, the authors state that their tool will address the “philosophical underpinnings of mixed methods,” and on 178-179 they state the need to prompt “researchers to report their philosophical or paradigmatic stance.” But isn’t this something that the experts should decide, rather than having the authors make prior assumptions about the centrality of this topic?

On line 200 they mention searching in the Journal of Mixed Methods Research, and I would add the International Journal of Multiple Research Approaches.

On line 267 the authors state that they will conduct “a targeted literature search to identify

additional mixed methods experts.” I find this to be too vague, since it doesn’t say anything about which literature will be searched or what criteria will be used for selection. I have personally conducted several citation searches that being with “mixed method*” and the results are so widely scattered that they would be useless for locating experts in the field.

On lines 272-273 the authors say they will use snowball sampling to expand their list of experts. This is a useful strategy, but snowball sampling is highly sensitive to the initial set of “seeds” that are used. So, this reinforces the point that I just made above about the importance of an explicit strategy for selecting the first set of experts.

Line 303 states that the items will be evaluated “using a 5-point Likert scale from highly relevant to non-relevant.” In my experience, this will lead to a highly skewed set of results, because very few of the items will be “non-relevant” after the authors’ intense screening process. I suggest that the scale run from “Minor Relevance” to “Highest Relevance” and that the authors use a 10-point rating scale. The usual argument for using 5-point and shorter scales is that it reduces respondent “burden,” but that is not a major issue when working with a highly educated panel who have all volunteered to engage in a rating process.

On line 303 the rating scale is described as “highly relevant to non-relevant,” but on lines 320-321 it is described as “Strongly disagree” to “Strongly Agree.” Again, no one is like to strongly disagree with the relevance of these items. If items “pile up” at the top of the rating scale, then al most all of them will meet the authors’ criteria for consensus, which would lead to a very lengthy instrument. As above, I would recommend a 10-point scale, and then use scores of either 7-10 or 8-10 as a cut-off for consensus.

Line 323 I disagree about whether there is agreement on the figure 85% for consensus. One common joke among Delphi researchers is that “there is no consensus about consensus.” I personally would recommend that 85% be set as an aspirational goal, but that 75% after the second round be considered adequate.

On line 345, I have found that one of the most useful aspects of the comments made by the experts is to adjust the wording of items. Will this be possible?

In the section on lines 347 to 367 it is unclear whether the 10 people are each going to rate all 10 articles. If so, I would prefer to have a sub-sample of the panel rate a larger number of articles. You would have the same number of ratings if five people each rated 20 articles. The reason for this recommendation is that variability among the articles is at least as important as variability among the raters.

Line 360 a random sample of “mixed method studies published in the last three years” will have a population of about 3,000 articles. How will the authors “stratify” such a large population to get just 10 (or 20) matching articles? I also worry that relying on PubMed for half the sample frame will lead to an over-emphasize health-related topics. What about substituting Scopus instead?

Reviewer #2: The authors have addressed the comments of the reviewers. Yet, there are still some remaining issues. The main comment is about the timeline suggested. The authors are planning a search period of the literature until September 2024 (line 225), a recruitment period of experts in September 2024 (line 298), and start the Delphi phase in October 2024 (line 299). Considering that the results of the review will be used to develop the Delphi questionnaire, it does not seem feasible to complete all the steps of the review, and develop and validate the questionnaire within one month. If the authors have already completed the review and questionnaire, they should include the actual date of the search, and state that the review is completed in the protocol.

Other comments are listed below:

• Line 106: replace “GRAMMAS” by “GRAMMS”.

• Line 166: Spell out “MMAT” since the first time it appears in the text.

• Lines 182-184: The authors mentioned “The MMR-RHS checklist is tailored for rehabilitation and health sciences. While it may be adaptable to other fields, it might not fully capture the nuances of mixed methods research in other disciplines”. To help readers understand, the authors should add an example of nuances of mixed methods not captured in the MMR-RHS.

• Line 219: The authors mentioned using free keywords and controlled vocabulary (MeSH and EMTREE). Yet, it is not clear what controlled vocabulary could be used for this topic. As mentioned previously, if the review is completed, the authors could add the actual search strategy used.

• Lines 238-241: “A two-stage screening process will be employed, first: initial screening will focus on mixed methods-specific guidelines. Second, quantitative and qualitative checklists will be reviewed selectively, prioritizing those frequently cited in mixed methods literature or those that have significantly influenced reporting practices”. This description of the screening process is different from what is usually found in reviews (i.e., first, screen titles and abstracts and then full-text papers). The process suggested will involve first identifying guidelines and then checklists. This could be done at the same time (i.e., identifying guidelines and checklist from titles/abstracts and then from full-text papers).

• Lines 241-243: “We will limit our in-depth review to the top 10 most relevant checklists for each methodology (quantitative and qualitative) based on citations.” It is not clear why the authors will limit to the top 10 checklists. Also, using number of citations to judge which ones are the “most relevant” can also be misleading. For example, more recent checklists are likely to be less cited but can be more relevant compared to a checklist developed in the 80s.

• Line 282: “This will be supplemented by a targeted literature search to identify additional mixed methods experts.” The authors should describe how they will process (e.g., where will they find the literature, how will they select the literature, etc.).

• Lines 381-388: “These studies will be systematically selected to represent a range of disciplines and reporting quality. The ten articles will be identified through a systematic search in PubMed and PsycINFO for mixed method studies published in the last three years. Stratified random sampling of the ten articles will be conducted to ensure representation across disciplines.” How will the authors assess the reporting quality of the 10 papers? Also, considering the latter sentence (random sampling), it seems that the authors will first identify all published mixed methods studies from the past 3 years indexed in PubMed and PsycINFO. Then, they will appraise the reporting quality of all identified mixed methods studies and classify them into disciplines. They will then randomly select 10 based on disciplines and reporting quality. This seems to be very time consuming. For example, only in Medline, there are more than 7000 papers published since 2022 that have used mixed methods in their titles.

• Reference #1 refers to two different books by Creswell (Designing and conducting mixed method research VS Research design: Qualitative, quantitative and mixed methods approaches). The authors should check which want they wanted to cite.

• Reference #5 is incomplete. Please check.

• Figure 1: There is a missing word at the end of step 6.

• The authors should avoid using contractions in the manuscript (e.g., there’s (line 81), It’s (line 118), …).

7. PLOS authors have the option to publish the peer review history of their article (what does this mean? ). If published, this will include your full peer review and any attached files.

**Do you want your identity to be public for this peer review?** For information about this choice, including consent withdrawal, please see our Privacy Policy .

Reviewer #1: **Yes: ** David L. Morgan

Reviewer #2: No

---

## [Author Response · Author response to Decision Letter 2]

2 Dec 2024

Nov 30th, 2024

The Editor-in-Chief

PLOS ONE

Dear Dr. Fàbregues

Revision of Manuscript

PONE-D-24-20917R1

Protocol for Developing a COnsolidated Checklist for Reporting MIXed-Methods Research (CORMIX) Using Modified Delphi.

The authors of the above-named manuscript would like to thank you for your e-mail dated 17 Oct 2024, containing the second review with constructive comments for the manuscript. The authors very much appreciate the comments and have revised the manuscript accordingly as attached. We are positive that the comments helped in enhancing the quality and scientific merit of the paper.

Below are separate pages, detailing our responses to the comments of the journal requirements and reviewer.

Please note that changes are made using “track changes” tool within the revised manuscript.

Thank you to the editor and the reviewer for the time spent on this manuscript. We look forward to receiving your kind response.

Sincerely yours,

The Corresponding Author

All in all, we thank you for your great review which we believe would add great value to the adjusted manuscript.

Detailed Responses to the Comments of the Journal Requirements, Editor and Reviewers

Comments from the Editor:

- Provide a rationale for the feasibility of the review and questionnaire development according to the stated timetable and clarify whether these stages have been completed.

We first thank the editor for their valuable comments and review of this manuscript. We have already completed several key phases of the project. The scoping review has been completed, yielding 50 relevant articles from an initial screening of 4,978 articles. From these articles, we have completed data extraction and synthesis, resulting in a preliminary list of items for the Delphi phase (provided in supplementary file). Additionally, the questionnaire for the Delphi phase has been developed and validated by the research team. Expert recruitment is currently ongoing, and based on our progress and existing commitments from experts, we anticipate completing expert recruitment by end of November, with the first Delphi round beginning in December 2024, followed by subsequent rounds through February-March 2025. This timeline is feasible given the substantial groundwork already completed and allows adequate time for expert engagement between rounds.

- On line 94 you say: “Attempts have been made to formulate reporting guidelines for mixed methods research to improve quality, but comprehensive standards are still lacking”. What do you mean by comprehensive? Please explain further why the published reporting frameworks are not comprehensive.

Thank you for your thoughtful comment. We acknowledge this statement was imprecise and have revised it for clarity. Rather than suggesting existing guidelines lack comprehensiveness, our intent was to highlight the need for a unified reporting tool that addresses both comprehensiveness and practical implementation challenges. As shown in our manuscript, existing tools like GRAMMS and MMARS have made valuable contributions but present specific practical limitations. For instance, MMARS requires users to cross-reference between multiple documents (JARS-Qual and JARS-Quant), which can complicate the reporting process and potentially compromise adherence to the tool. Similarly, while MMR-RHS provides robust guidance, its focus on rehabilitation and health sciences may not fully capture reporting needs across different disciplines. We have revised the text to better reflect these nuances, now reading: "While robust guidelines exist for mixed methods research reporting, there remains a need for a unified, standalone checklist that streamlines the reporting process and provides detailed operational guidance across disciplines."

- On line 106, when discussing the JARS, you state: “While more comprehensive than GRAMMAS, it does not provide the level of detail some researchers might need for comprehensive reporting across all aspects of a mixed methods study”. Please explain this statement further, as we believe that the JARS is quite detailed.

We appreciate this comment and agree that our original statement didn't accurately reflect JARS's detailed nature. JARS-MMARS indeed provides comprehensive guidance for reporting mixed methods research. Our concern was not with its level of detail but rather with practical implementation challenges. We have revised this section to better reflect this distinction. The revised text now reads: "While more comprehensive than GRAMMS, it was primarily developed for psychology and related fields, while mixed methods research spans multiple disciplines including healthcare, education, and social sciences. Additionally, users must cross-reference between multiple documents (JARS-Qual and JARS-Quant), which can complicate the reporting process and compromise adhering to the tool. A standalone, unified checklist could enhance usability and hence, improve reporting quality."

- On line 124, you state that the tool will address “the philosophical underpinnings of mixed methods”. What are these underpinnings? How will they be addressed?

Thank you for this important question. We acknowledge that our statement about "philosophical underpinnings of mixed methods" needs clarification. We have revised this section to be more precise about our approach. Rather than presuming specific philosophical underpinnings that must be reported, our tool will be guided by expert consensus on what aspects of philosophical and paradigmatic considerations, if any, should be reported in mixed methods research. Through our completed literature review, we found that philosophical and paradigmatic considerations are frequently discussed in mixed methods literature. However, we recognize that the decision about whether and how these should be incorporated into reporting guidelines should emerge from expert consensus during our Delphi process rather than being predetermined. Therefore, the role and importance of reporting such considerations will be evaluated through expert consensus in our study. We have amended the manuscript to reflect this more accurately, removing any implications that we have predetermined what philosophical elements should be included in the reporting guidelines.

- Include the full search strategy used or planned as a supplementary file.

We included the full search strategy in the supplementary file.

- Clarify the issues raised by reviewer 2 regarding the screening process and the selection of checklists for review.

We appreciate the reviewer's comment about the screening process and checklist selection. We have revised the manuscript to clarify both aspects. The screening process followed standard systematic review procedures: first, titles and abstracts were screened by two independent researchers using Rayyan, followed by full-text screening of potentially relevant articles. Our search identified 13,439 articles, which after duplicate removal resulted in 4,978 articles for title/abstract screening. Full-text screening was completed for 121 articles, with 50 articles ultimately included.

Regarding the selection of quantitative and qualitative checklists, this was conducted as a supplementary search to complement our primary focus on mixed methods-specific reporting guidance. Any potential limitations from focusing on frequently cited checklists were mitigated through our comprehensive methodology, which includes expert input through the Delphi process to ensure no crucial reporting elements are overlooked.

- If the review has been completed, please indicate the implications of not following the reviewers’ suggestions and how these issues will be addressed. We believe that the reviewers’ comments are important enough to be considered.

We thank the editor and reviewers for their valuable feedback. Although the review has been completed, we have carefully considered the reviewers' comments and addressed them in several ways: First, we have clarified our screening methodology in the manuscript to accurately reflect the systematic process we followed: initial title/abstract screening of 4,978 articles by two independent researchers, followed by full-text review of 121 articles, resulting in 50 included articles. This aligns with standard systematic review procedures. Regarding the selection of quantitative and qualitative checklists, while we initially used citation counts as one selection criterion, we recognize this could potentially miss relevant newer guidelines. However, our methodology incorporates several safeguards against overlooking important reporting elements: the comprehensive capture of items from mixed methods literature, the expertise of our Delphi panel, and the opportunity for experts to suggest additional items or sources during the Delphi process. The Delphi process will allow us to validate and potentially expand upon our initial findings, ensuring no crucial reporting elements are missed regardless of their source.

- Provide more details on how mixed methods experts will be identified, both in terms of the literature to be searched and the criteria for selecting experts.

We have expanded our description of expert identification to detail both our search strategy and selection criteria. We added more details into the text. “Expert identification will involve a multi-faceted approach to ensure comprehensive representation across disciplines. For this study, experts are defined as individuals with demonstrated knowledge and skills in mixed methods research. The selection criteria is defined in Table 1. We will begin by extracting author information from our scoping review of mixed methods literature. This will be supplemented by a targeted literature search in Scopus and Web of Science using terms such as 'mixed methods research' and 'mixed methodology' to identify authors who have published extensively in this field. We will focus on authors who have published multiple mixed methods studies or methodological papers about mixed methods research. Authors' institutional profiles and Google Scholar profiles will also be reviewed to confirm their ongoing engagement with mixed methods research. The lead author will compile an initial list of experts, categorized by research discipline, which will be reviewed and supplemented by the full research team to ensure comprehensive representation. The experts' biographies and publication records will be consulted to verify their research design expertise, including their research interests, courses taught, and scientific publications. Furthermore, we will contact the Mixed Methods International Research Association and other relevant organizations to nominate experts based on pre-determined selection criteria (Table 1). To address disciplinary differences, we will strive for a balanced representation of experts across various fields, including but not limited to psychology, sociology, health sciences, and education.”

- Consider reviewer 1’s suggestions regarding measurement tools, criteria for agreement, and other methodological issues.

Upon the discussion with the team we came to agree with the reviewer and adjusted our questionnaire to address the reviewer’s comments. We thank the reviewer for these methodological suggestions and have made several specific adjustments to strengthen our measurement approach. We have revised our consensus criteria and measurement tools as follows: We now use a 10-point Likert scale instead of the originally proposed 5-point scale, allowing for more nuanced expert assessment of items. We have also modified our consensus thresholds, using ≥85% agreement in the first round (defined as scores 7-10 on the Likert scale), followed by a 75% threshold in subsequent rounds. Additionally, we have added a separate 3-point scale for rating item clarity (1- completely unclear, 2- somewhat clear, and 3 – completely clear) to ensure both item retention and clarity are systematically assessed. Each item is also followed by an open-text field allowing experts to provide detailed feedback and justification for their ratings.

- Clarify how the 10 articles for the pilot will be selected.

We appreciate this comment and have refined our pilot testing approach. We have increased the number of articles from 10 to 15 to ensure broader representation across disciplines, and have clarified our selection strategy. The pilot articles will be selected through a structured process: First, we will conduct a search in Scopus and PsycINFO to identify mixed methods studies published in the last three years. To ensure disciplinary diversity, we will stratify our selection to include articles from four major disciplines (psychology, sociology, health sciences, and education), aiming for 3-4 articles from each field where feasible. We will use convenience sampling within each disciplinary stratum, selecting articles that demonstrate varied approaches to mixed methods research. This balanced representation across disciplines will allow us to assess the checklist's applicability and usefulness across different research contexts and methodological traditions. The same set of articles will be reviewed by all members of the pilot team to enable consistent assessment of the checklist's reliability and usability.

- Respond to any other issues raised by the reviewers.

We addressed all comments raised by the reviewers.

- There are several typographical and grammatical errors throughout the paper. These need to be corrected: e.g. lines 64-65, 101-102.

We thank the editor again and we reviewed the manuscripts for typographical and grammatical errors.

----

Reviewer #1: A this point, I still have a number of comments, most of which are rather small.

On lines 117-118, the authors state that their tool will address the “philosophical underpinnings of mixed methods,” and on 178-179 they state the need to prompt “researchers to report their philosophical or paradigmatic stance.” But isn’t this something that the experts should decide, rather than having the authors make prior assumptions about the centrality of this topic?

Thank you for your comment regarding the philosophical underpinnings of mixed methods research. We agree that the importance of reporting philosophical and paradigmatic stances should ultimately be determined by the expert consensus process rather than by prior assumptions from the authors. The literature review and scoping review we conducted suggest that experts may consider these aspects important to report, but you rightly point out that this should be confirmed through the Delphi process. To address this, we have amended the manuscript to clarify that the expert panel will be responsible for deciding which elements, including philosophical and paradigmatic considerations, are essential to include in the CORMIX reporting guideline. The revised wording acknowledges the key role of expert consensus in determining the final checklist items, while still noting that our preliminary work indicates these aspects could be relevant for discussion. We appreciate you raising this important point and prompting us to refine our language around the reporting philosophical stances. Ultimately, the expert consensus process will be the arbiter of which items are included in CORMIX.

On line 200 they mention searching in the Journal of Mixed Methods Research, and I would add the International Journal of Multiple Research Approaches.

Thank you for suggesting the addition of the International Journal of Multiple Research Approaches to our search strategy. We appreciate you bringing this relevant journal to our attention. Since receiving your feedback, we have completed a separate search of this journal to ensure comprehensive coverage of the literature. It's important to note that the International Journal of Multiple Research Approaches is indexed in databases we had already searched, including PubMed and ProQuest. Therefore, our original search strategy likely captured articles from this journal. However, to be explicit and thorough, we have now included the International Journal of Multiple Research Approaches in the list of journals specifically searched, as detailed in the revised manuscript.

On line 267 the authors state

---

## [Editor Report · Decision Letter 2]

27 Dec 2024

PONE-D-24-20917R2Protocol for Developing a COnsolidated Checklist for Reporting MIXed Methods Research (CORMIX) Using Modified DelphiPLOS ONE

Dear Dr. Jaam,

Thank you for submitting your manuscript to PLOS ONE. After careful consideration, we feel that it has merit but does not fully meet PLOS ONE’s publication criteria as it currently stands. Therefore, we invite you to submit a revised version of the manuscript that addresses the points raised during the review process.

We look forward to receiving your revised manuscript.

Kind regards,

Sergi Fàbregues

Academic Editor

PLOS ONE

Journal Requirements:

Additional Editor Comments:

While the authors have addressed most of the reviewers’ comments, the reviewers note that a few minor issues still need to be addressed. These issues, with which I agree, include:

- Clarify how all the steps of the review will be completed in the short period of time mentioned between the date of the search and the start of data collection.

- Correct several typos and add some examples, as suggested by the reviewers.

- Clarify and add additional information about the methods employed (especially regarding the search process for identifying experts) and their limitations.

- Correct several problems in the list of references.

- In the response to the reviewers letter, provide a justification for procedures and steps that were performed in a different way than suggested by the reviewers.

---

## [Author Response · Author response to Decision Letter 3]

19 Jan 2025

Jan 19th , 2025

The Editor-in-Chief

PLOS ONE

Dear Dr. Fàbregues

Revision of Manuscript

PONE-D-24-20917R2

Protocol for Developing a COnsolidated Checklist for Reporting MIXed-Methods Research (CORMIX) Using Modified Delphi.

Thank you for your additional feedback on our manuscript. We appreciate the opportunity to address the points raised by the editor. We wish to highlight that we only received the comments shared by the editor without new comments from the reviewers. We are positive that the comments helped in enhancing the quality and scientific merit of the paper.

Below are separate pages, detailing our responses to the comments of the journal requirements and editor.

Please note that changes are made using “track changes” tool within the revised manuscript.

Thank you to the editor for the time spent on this manuscript. We look forward to receiving your kind response.

Sincerely yours,

The Corresponding Author

All in all, we thank you for your great review which we believe would add great value to the adjusted manuscript.

Detailed Responses to the Comments of the Journal Requirements and Editor

Journal Requirements:

We have reviewed each reference and manually adjusted any inconsistencies. No retracted paper was cited in this research. All changes were made using track changes. These are the changes in brief:

Journal names were all abbreviated. Edition and volume were added in few missing references. Standardized formatting for references, in terms of punctuation and doi links.

Additional Editor Comments:

While the authors have addressed most of the reviewers’ comments, the reviewers note that a few minor issues still need to be addressed. These issues, with which I agree, include:

- Clarify how all the steps of the review will be completed in the short period of time mentioned between the date of the search and the start of data collection.

We confirm that several key phases have already been completed. (1) Scoping review (completed September 2024): 50 relevant articles identified from 4,978 screened articles. (2) Questionnaire development and validation (completed October 2024). (3) Expert recruitment (began December 2024, currently ongoing)

To better illustrate this progress, we have added Figure 3 showing completed steps and timeline under a new section “Completed steps" in the manuscript

- Correct several typos and add some examples, as suggested by the reviewers.

We conducted a thorough review of the manuscript and corrected all typos. In the previous review, we added an example related to philosophical stances in mixed methods research “For instance, in social sciences, philosophical stances (e.g., pragmatism, constructivism, transformative paradigms) often fundamentally guide research design, yet the MMR-RHS doesn't explicitly prompt reporting of these considerations.”

- Clarify and add additional information about the methods employed (especially regarding the search process for identifying experts) and their limitations.

Based on the previous review, we have expanded our description of expert identification methods to include detailed selection criteria in Table 1, systematic literature searches in Scopus and Web of Science, professional network outreach through MMIRA, snowball sampling from identified experts. We also added a paragraph acknowledging the limitations of our approach. “We acknowledge several limitations in our expert identification strategy. First, reliance on academic publications may underrepresent practitioners and experts who primarily contribute through other channels. Second, English-language dominance in major databases could limit identification of experts from non-English speaking regions. Third, citation metrics and publication counts, while useful indicators, may not fully capture expertise, particularly for emerging scholars or those focusing on specific disciplinary applications of mixed methods. To mitigate these limitations, we will actively seek recommendations from professional organizations and use snowball sampling to identify experts who might be missed through traditional academic metrics. We will also make specific efforts to include experts from diverse geographical regions and academic traditions.”

- Correct several problems in the list of references.

We have reviewed and updated all references.

- In the response to the reviewers letter, provide a justification for procedures and steps that were performed in a different way than suggested by the reviewers

We thank the editor for their comment. We have addressed and agreed with almost all comments raised by the reviewers in the previous review rounds. To summarize these were the changes and their justifications:

The reviewer suggested using a 10-point Likert scale instead of our originally proposed 5-point scale. We agreed with this suggestion and implemented it, as a 10-point scale offers better discriminatory power when working with expert panels. Additionally, we modified our consensus thresholds to ≥85% agreement (scores 7-10) and subsequent rounds: ≥75% agreement. This modification allows for more nuanced expert assessment while maintaining rigorous consensus standards.

We modified our database selection from PubMed to Scopus and PsycINFO as suggested in previous rounds of review as it provides broader disciplinary coverage, reduces potential bias toward health sciences and better captures social science and education research allowing more comprehensive identification of mixed methods studies.

While maintaining the suggested snowball sampling approach, we expanded our expert identification strategy to include systematic database searches in Scopus and Web of Science, engagement with professional organizations (MMIRA), and review of institutional profiles and Google Scholar metrics. This comprehensive approach helps ensure a diverse, qualified expert panel while addressing potential limitations of any single identification method.

While a reviewer suggested having fewer people rate more articles for the pilot of the tool, we opted for a different approach: Increased the number of articles from 10 to 15 and maintained the full pilot team of 10 members rating all articles. This approach provides broader disciplinary representation, and greater reliability in assessing the checklist's usability through better assessment of inter-rater consistency.

We believe we have taken all reviewers comments raised in previous rounds into consideration. These modifications from the original suggestions were made to enhance the scientific rigor of our study while maintaining feasibility and ensuring comprehensive representation across disciplines. Each decision was made after careful consideration of the reviewers' feedback and practical constraints of the study.

---

## [Editor Report · Decision Letter 3]

4 Feb 2025

PONE-D-24-20917R3Protocol for Developing a COnsolidated Checklist for Reporting MIXed Methods Research (CORMIX) Using Modified DelphiPLOS ONE

Dear Dr. Jaam,

Thank you for submitting your manuscript to PLOS ONE. After careful consideration, we feel that it has merit but does not fully meet PLOS ONE’s publication criteria as it currently stands. Therefore, we invite you to submit a revised version of the manuscript that addresses the points raised during the review process. The reviewer reports are attached. Please provide a point-by-point response to the reviewers' comments. Many thanks.

We look forward to receiving your revised manuscript.

Kind regards,

Sergi Fàbregues

Academic Editor

PLOS ONE

Journal Requirements:

Additional Editor Comments:

The Revision 2 reviewer reports are attached. Please provide a point-by-point response to the reviewers' comments. Many thanks. Sergi

---

## [Author Response · Author response to Decision Letter 4]

2 Mar 2025

Mar 2nd, 2025

The Editor-in-Chief

PLOS ONE

Dear Dr. Fàbregues

Revision of Manuscript

PONE-D-24-20917R3

Protocol for Developing a COnsolidated Checklist for Reporting MIXed-Methods Research (CORMIX) Using Modified Delphi.

Thank you for sharing the reviewer’s comments file. However, we have already addressed and responded to these comments in second round of reviews.

I also wish to clarify the following timeline for the review process, I have attached the corresponding files for each.

Review 1: comments received in 24 July 20204, addressed and submitted on 3rd of September, 2024 (File: Review 1)

Review 2: comments received on 17 Oct 2024, addressed and submitted on 30 November, 2024 (File: Review 2)

Review 3: comments received on 27 December 2024, addressed and submitted on 19th January, 2025 (included editor’s comments only) (File: Review 3)

Review 4: comments received on 4th of February 2025 (which were the same comments received and addressed in review 2) (File Review 4 – with no responses as they are the same as File Review 2)

Please let us know if there are new review comments that we need to address further as the ones shared were from a past round.

Thank you to the editor for the time spent on this manuscript. We look forward to receiving your kind response.

Sincerely yours,

The Corresponding Author

---

## [Editor Report · Decision Letter 4]

9 Mar 2025

Protocol for Developing a COnsolidated Checklist for Reporting MIXed Methods Research (CORMIX) Using Modified Delphi

PONE-D-24-20917R4

Dear Dr. Jaam,

We’re pleased to inform you that your manuscript has been judged scientifically suitable for publication and will be formally accepted for publication once it meets all outstanding technical requirements.

Kind regards,

Sergi Fàbregues

Academic Editor

PLOS ONE
---

## [Editor Report · Acceptance letter]

PONE-D-24-20917R4

PLOS ONE

Dear Dr. Jaam,

I'm pleased to inform you that your manuscript has been deemed suitable for publication in PLOS ONE. Congratulations! Your manuscript is now being handed over to our production team.

Kind regards,

on behalf of

Dr. Sergi Fàbregues

Academic Editor

PLOS ONE